# A Music Therapy Intervention for Refugee Children and Adolescents in Schools: A Process Evaluation Using a Mixed Method Design

**DOI:** 10.3390/children9101434

**Published:** 2022-09-21

**Authors:** Evelyn Heynen, Vivian Bruls, Sander van Goor, Ron Pat-El, Tineke Schoot, Susan van Hooren

**Affiliations:** 1Faculty of Psychology, Department of Clinical Psychology, Open University of The Netherlands, 6419 AT Heerlen, The Netherlands; 2Zuyd University of Applied Sciences, 6419 DJ Heerlen, The Netherlands; 3Safe & Sound, 6217 LD Maastricht, The Netherlands; 4Faculty of Psychology, Department of Methods and Statistics, Open University of The Netherlands, 6419 AT Heerlen, The Netherlands

**Keywords:** music therapy, refugee children/adolescents, process evaluation, sense of belonging, affect, resilience

## Abstract

Refugee children and adolescents have often experienced negative or traumatic events, which are associated with stress and mental health problems. A specific music therapy intervention is developed for this group in school settings. The aim of the present study was to set the first steps in the implementation of this intervention. A process evaluation was performed using a mixed method design among refugee children and adolescents (6–17 years) at three different schools in the Netherlands. Interviews were conducted with teachers and music therapists before, at the midpoint, and after the intervention. At these moments, children completed a classroom climate questionnaire and a visual analogue scale on affect. The results indicate that the intervention strengthens the process of social connectedness, resulting in a “sense of belonging”. The intervention may stimulate inclusiveness and cultural sensitivity, and may contribute to a safe environment and the ability of teachers to adapt to the specific needs of refugee children. Refugee children and adolescents showed a decrease of negative affect during the intervention. When implementing the intervention in schools, it is important to take into account the initial situation, the prerequisites for the intervention, the professional competence, the experience of music therapists, and the collaboration and communication between the professionals involved.

## 1. Introduction

In the past decade, the rising number of people entering Europe and the USA in search of safety has captured the world’s attention. Up to 50% of them are children and adolescents, with a total of 25,000 unaccompanied minors applying for asylum annually across 80 countries [1]. Most of them have negative and traumatic experiences in their home country, while travelling to Europe, and during the search for a safe new home. They lost their social network, have to deal with uncertainties surrounding their future, and may experience cumulative stress of forced migration. This has shown to be strongly related with stress and mental health disorders [2,3], such as post-traumatic stress disorder, depression, anxiety, and substance use disorders [3,4,5,6,7,8]. Untreated, these disorders may become chronic and undermine functioning [9,10,11]. 

In schools, stress and instability is also recognized in refugee children, as for example seen in fewer social relationships and less positive integration and participation [12], resulting in a higher school drop-out compared with non-refugee children. Refugee children show a higher rate of concentration problems, more aggressive incidents, anxiety, and worry more about their own safety and safety of others [4,12]. This behavior has been shown to influence classroom climate and the relationship between children and teachers [13]. Proper programs to prevent or address these problems are urgently needed.

Due to their daily contact with the children, schools obtained a growing role in helping refugee children and adolescents in preventing or adapting their (mental health) needs [14]. Positive education can prevent children from psychosocial problems and can improve learning and integration, strengthen psychological well-being, and prevent children from the development of mental health problems [14]. In order to address mental health of refugee children in schools, it is necessary to use specific programs that are sensitive for culture, take language difficulties into account, and have a solid evidence base. There is actually only a small amount of research focusing on interventions in refugee children that promote mental health or moderate stress in terms of resilience [15,16]. In fact, given the large number of children and adolescents displaced by war, there are regrettably few treatment studies available, and many of them were of low methodological quality [17].

Non-verbal interventions such as music therapy have shown to improve a positive psychosocial development of the child, reduce stress, and strengthen their resilience [18]. The use of music can serve to bridge the gap between languages and cultures, since it is universal to all cultures [19]. Neuroscientific studies have shown that music decreases physiological arousal and modulates activity in brain structures that are involved in emotional and motivational processes [20,21]. It is assumed that methodologically using music in therapy can strengthen the impact of music. Music therapy gives opportunities for self-expression and strengthening their (ethnic) identity [22,23]. In a group, music therapy provides opportunities for sharing and communicating on beliefs and hope for the future. In this line, it promotes peer-support and social engagement [24]. The intervention “Safe & Sound” is a music therapy intervention for children and adolescents who experienced negative and traumatic events during war and flight [18,25]. The intervention aims to focus on aspects such as helping each other, working together, feelings of safety, and collaborative learning. In practice, the intervention is already used in Dutch primary schools as a prevention strategy. Studies on the effectiveness of interventions in this population are sparse. This study aimed to set the first steps in the implementation of the intervention and in investigating the perceived effects of a music therapy intervention in refugee children. A process evaluation was conducted in order to obtain insight in the process and process results of the intervention “Safe & Sound” in general and on affect and the learning climate experienced by the refugee children and adolescents. A second aim was to identify influencing factors for the implementation of the intervention in schools. In addition, we wanted to evaluate whether the intervention was conducted as intended (treatment integrity). 

## 2. Methods

### 2.1. Design

The current research was conducted as a mixed-methods (embedded design) longitudinal study, in which both quantitative as well as qualitative methods were applied [26]. A process analysis was conducted in three different schools in the Netherlands. At three measurement moments, interviews were conducted with teachers and music therapists and questionnaires were filled in by the children and adolescents: before the start of the intervention after the summer holidays (T_0_), at the midway-point of the intervention in autumn (after one month, T_1_), and after completion of the intervention before the Christmas holidays (after three month T_2_). In addition, after each session, music therapists answered questions on treatment integrity.

The schools consisted of two elementary schools and one secondary school. These schools have specific classes with educational programs for non-native children and adolescents (most of them are refugees and asylum seekers). The educational programs have the aim to help them integrate and learn the new language and regular knowledge and skills. Schools were recruited by the professional network of the music therapist who developed the intervention. During recruitment, schools were informed about the topic and the method of the research. All schools gave consent to participate in the music therapy sessions and the present study. An information meeting was held after consent with all participating teachers at each school. 

### 2.2. Participants

The participants consisted of refugee children and adolescents even as their music therapists and teachers. The children and adolescents were non-native, between the age of 6 to 17 years following education at elementary or secondary schools (for more information see results Section 3.1). All teachers involved in the class for non-natives were asked to participate. One of the music therapists was the developer of the intervention “Safe & Sound (SG)”. Two other music therapists were recruited by a formal application procedure. Inclusion criteria for music therapists were a bachelor’s degree in music therapy and at least three years of experience with children or adolescents and/or trauma-related problems. In addition, these therapists received an additional two-day training on trauma and resilience from the music therapist who developed the intervention. Before participation, informed consent was obtained from children, their parents, teachers, and music therapists. 

### 2.3. Music Therapy Intervention “Safe & Sound”

The music therapy intervention “Safe & Sound” was developed in order to strengthen resilience and self-control of children/adolescents who grow up under difficult circumstances such as refugees. In addition, the purpose of the intervention was to prevent or decrease psychosocial complaints and problems which result from a stressful or traumatic event in the past. 

The intervention “Safe & Sound” includes two key elements, a classroom-based intervention and an individual intervention. The classroom sessions are embedded in the educational program of the participating schools. During the present study, classroom sessions consisted of ten sessions with a maximum duration of one hour per session. During these sessions, the music therapist worked on a positive climate in the group and aimed to optimize the prerequisites for the learning process, i.e., feeling safe and relaxed and open for new (learning) experiences. During the sessions, the therapist and children work together on interpersonal goals, such as listening to each other, helping each other, and trusting each other. Each classroom session has a specific theme (e.g., making new friends, sharing stories, dealing with difficult situations and emotions, and their talents), which were also based on topics that children want to share with each other or on characteristics of the atmosphere in the group. Finally, children/adolescents of the class present the song they worked out through the sessions to their teachers and parents/care givers, who are stimulated to discuss the song with their child. 

During group sessions, music therapists are vigilant for children/adolescents who display behavioral issues or are at risk for further development of (trauma related) psychosocial problems. Music therapists are trained to signal these issues and will discuss this with the teacher of the group, the parents/care givers, school psychologist, or school doctor. If there are indications for further support of the psychosocial development of the child, individual sessions of “Safe & Sound” can be indicated. These individual sessions focus on the individual needs and possibilities of the child/adolescent and can thus provide attention and support to the psychosocial development of the child/adolescents. Those individual sessions mainly focus on stabilization and further try to set important first steps in the treatment of trauma-related problems, always in collaboration with the personal environment of the child/adolescent. If there are signals for need of further treatment, this will be discussed with the child/adolescent family, (remedial)teachers, and school doctor/psychologist. 

During the time of the intervention, there was supervision and intervision for the music therapists held by the developer of “Safe & sound (SG)” during two group and two individual sessions. 

### 2.4. Procedure

Directly after the start of the class year 2019/2020, parents and children/adolescents were informed by teachers about the goal and the procedure of the intervention and the study. The intervention “Safe & sound” was offered to all children/adolescents in the participating classes. Children/adolescents of one class participate together in a session one hour each week within a fixed schedule. If children/adolescents were considered for the individual sessions, parents were informed and asked for their approval. Individual sessions were indicated when teachers report psychological (anger, stress, anxiety, sadness) or physical (sleeplessness, pain) problems. Children/adolescents received questionnaires at T_0_, T_1_, and T_2_. Furthermore, music therapists and teachers were interviewed at T_0_, T_1_, and T_2_. After every music therapy session (both classroom sessions and individual sessions), music-therapists completed a questionnaire on treatment integrity (see Figure 1). 

### 2.5. Data Collection Methods

#### 2.5.1. Interviews 

Semi-structured interviews were conducted with teachers and music therapists to gain more insight in the process of embedding the intervention and its results in the participants’ perspective. Interviews focused on the group and individual part of the intervention. The interview questions were based on the guideline for process evaluations of Movisie [27]. The main topics for the teachers were appreciation, experiences, the scope, and the influencing factors for treatment (success and failure). Topics of interviews for music therapists focused on the execution of the intervention of “Safe & Sound” and its results. The interviews were audiotaped and transcribed for analyses.

#### 2.5.2. Measurement Instruments 

In order to evaluate the experiences of participating children and adolescents, a possible language barrier was taken into account by using simple language, conforming to the principles of “Easy language” [28]. In order to allow the participation of children and adolescents with different cultural backgrounds and a broad age range (6–17 years), we selected instruments with a visual attractive format and piloted the items and answer methods in a subsample of refugee children and adolescents. Based on this, we decided to limit the number of items, to include items on somatic complaints, and to include visuals, see Appendix A and Appendix B.

Visual analogue scale to measure positive and negative affect

The visual analogue scale (VAS) is a psychometric measurement method designed to document the characteristics of emotions and symptom severity. In the present study, children/adolescents were shown a horizontal line (10 cm) and asked to mark their level of positive and negative affect on the line ([29] VAS, Appendix A). On the left side is the minimum score (“*don’t feel the emotion at all*” = 0), and on the right is the maximum score (“*feel the emotion really strongly*” = 10). The score on the VAS is the number of centimeters (with one decimal) between the minimum score and the line indicated by the participant. A high score on the VAS means that the emotion is experienced to a high degree. Acceptable psychometric properties have been reported for a digital VAS for measuring anxiety [30]. In the present study, six items were investigated for negative affect (headache, stomach ache, angry, easily angered, sad, annoyed, anxious) and two for positive affect (happy, pride).

A CFA was conducted in R version 4.1.2 with the lavaan package, version 0.6–9 [31]. A correlated two-factor model with negative and positive affect as two correlated factors were modeled for each time point separately, which resulted in three CFAs. CFI, RMSEA, and SRMR were used as fit measures, with *CFA* > 0.90, *RMSEA* and *SRMR* < 0.08 as cutoff values for adequate model fit. The correlated two-factor models showed very poor fit (see Table 1). *CFI* was highest for T1 (*CFI* = 0.82), and substantially lower for the other time points. RMSEA and SRMR were all well above the threshold of 0.08. An inspection of the performance of each item showed that item “easily angered” loaded poorly (below 0.2) on negative affect at every time point. The removal of “easily angered” improved the fit somewhat, with SRMR getting closer to an adequate threshold, but overall, the model fits were poor. The poor fit is reflected in the low reliabilities of the scales. McDonald’s omega was low for positive affect (around 0.45), and just about acceptable for negative affect (between 0.63 and 0.68) [32]. Therefore, we only used negative affect in the subsequent analyses. 

Special Education Classroom Climate Inventory

The special education classroom climate inventory (SECCI, see Appendix B) was designed to assess the classroom climate in schools for special education, secure residential care, and youth prisons. In the present study, we only used six items to investigate overall classroom climate on six topics, i.e., support, growth, repression, atmosphere, environment, and safety. Items were rated by refugee children and adolescents using school grades according to the Dutch school system on a 10-point scale ranging from 10 = *outstanding* to 1 = *very poor*. The questionnaire has shown to be valid and reliable for use in Dutch special education [33]. 

### 2.6. Treatment Integrity 

A questionnaire was used to investigate treatment integrity and to show to what extent the intervention was implemented as intended. This questionnaire consisted of eight questions. Questions focus on: (1) the degree of how music therapists followed the instruction manual, (2) feelings of competence, and (3) applied reflective questions on feelings, emotions, and talents. Each question needed to be answered with a percentage between 0–100. This questionnaire was filled in after each group session and each individual session. Treatment integrity was considered sufficient when both therapist adherence and competence reached a percentage of >60% at group level [34].

### 2.7. Data Analyses 

Separate analyses were performed for the qualitative- and quantitative data. After that, the results were integrated in order to answer the research questions (see discussion).

#### 2.7.1. Qualitative Data Analysis

The interviews were analyzed by means of qualitative content analyses [35], according to the constant comparison analyses method (CCA) [36]. A process of inductive reasoning resulted in salient categories of meaning and relationships between categories. By means of the CCA, relevant data were deductively grouped in the category with the best fit. One researcher (VB) started inductively open coding of the teacher interview data. After reading the transcripts, text fragments that were relevant for the research question were marked and provided with an open code. Initially, “in vivo codes” were used, which entails the text fragments labeled in the words or short phrases of the respondent as far as possible. Subsequently, together with a second researcher (TS), these codes were compared and grouped in the category or subcategory of best fit.

Secondly, the qualitative data of the music therapists were analyzed. By means of a deductive approach, text fragments were coded, by making use of the already developed coding tree for the teachers. Several subcategories were added to the coding tree. During the process of analyzing the codes, categories and subcategories were constantly compared with each other to be refined. Furthermore, memos were used considering the research process and the content. During the analyses, the research questions were demarcated to the intervention on group level. The reason is that the interview data provided insufficient insight in the process of the individual part of the music intervention “Safe & Sound”. 

#### 2.7.2. Quantitative Data Analysis

The questionnaires of the children were analyzed using hierarchical multilevel regression modelling in order to investigate differences over time with regard to negative affect of the child or adolescents (VAS) and the climate in the class (SECCI). Analyses were performed in R version 4.1.2 [37]. The mice-package [38] was used for missing value imputation. The nlme-package, version 3.1–153 [39], was used to conduct the multilevel regression analyses. Time points were nested within individuals. Negative affect was modeled as a criterion variable, as were the six SECCI variables: support, growth, atmosphere, environment, repression, and safety. For each criterion variable (eight in total), four sequential multilevel models were tested: (1) the ‘null’ model, (2) a growth model with random intercept-only, (3) a growth model with random intercept and slope, and (4) a model including school-type as a predictor. All models were fitted using maximum likelihood estimation. To decide upon the best model, a combination of indicators was used. The AIC of models were compared. Lower AIC indicates a better model fit, and as such, models with the lowest AIC were preferred. In addition, the difference in deviance (−2LL) between nested models were tested with a chi-squared test with the difference in degrees of freedom (df) between the models as the chi-squared df. A significant difference in deviance combined with a lower AIC than the compared models was used to determine if a more complex model of two compared models could be preferred. The scores on the treatment integrity items were analyzed using descriptive statistics. In calculating mean scores, we only used the scores of the music therapists who did not apply the intervention previously. By excluding the scores of the music therapist who developed the intervention, we prevented a response bias.

### 2.8. Trustworthiness

The credibility of the qualitative analysis was ensured by using different sources of information concerning the same events: the interviews with music therapists and teachers in different schools (*data triangulation*). In addition, throughout the analysis of the qualitative data, two different authors reflected on the research process to identify areas for further investigation and analysis (*investigator triangulation*).

To validate the outcomes, the results of the process of analysis were discussed with the research team, including one music therapist (SvG) who functioned as the linking pin between participating music therapists and teachers (*member check*). Furthermore, credibility was sought by using multiple methods of data collection, both qualitative and quantitative (*methodological triangulation* [40]). One of the authors (TS) provided guidance on whether the analysis of the qualitative data was in line with accepted standards (*dependability*) and supervised the analysis process of the qualitative data and transcripts for accuracy (*confirmability*). 

Throughout the research process, meetings on a regular basis reviewed the scientific and organizational aspects of the project (*peer debriefing*). Finally, *transferability* was ensured by providing descriptive data of the study context (*thick description*) to enable readers to evaluate whether the findings are transferable to other care contexts.

## 3. Results

### 3.1. Participants

108 children and adolescents were included. Their age ranged from 6 to 17 years (*Mean* = 11.63, *SD* = 2.79). 50 boys and 50 girls were included; for eight participants, gender information was missing. The children and adolescents came from 35 different countries, most of them from Syria (*N* = 28), Iran (*N* = 11), and Eritrea (*N* = 8). At the time of the start of the intervention, 47 children lived in their own home and another 28 in an asylum center; the information of 33 children on their living situation was missing. In addition, all teachers (*N* = 7) involved and all music therapists (*N* = 3) that applied the intervention were interviewed.

### 3.2. Qualitative Results 

Results of the qualitative analyses indicated that the intervention strengthens the process of social connectedness, resulting in a “sense of belonging” (Figure 2). The *process of social connectedness* and the results of this process was influenced by several factors (Figure 2), which will be discussed further. 

#### 3.2.1. Strengthening the Process of Social Connectedness 

The intervention “Safe & Sound” has shown to strengthen the *process of social connectedness*, which is referred to as the experience of belonging and relatedness between people. Social connectedness can be based on the present results, defined as the process of social bonding between the children/adolescents during time. Both teachers and music therapists report social connectedness in terms of breaking out of their shell, communicating and sharing emotions, listening to each other, discovering each other’s talents, involvement in relationships, and sharing positive experiences. A participating teacher reported: “*That new girl at (says teacher’s name) class, yes, she then immediately joins (says student’s name) and (says student’s name) and yes she doesn’t really want to leave there*.” Another teacher said: “*I think learning to listen to each other, so learning to listen to what someone else is doing or saying or playing, that is a very good skill*”.

These social interactions were visible between all children/adolescents of the group, irrespective of culture, language, age, gender, behavior, and whether or not they are new in the group. This suggests that the intervention may stimulate inclusiveness at school. A teacher reported: “*In this group, there are children who have more problems or display more disturbing behavior. But I see that those children are not only included by the therapist, but also by other children. It looks like they care for each other and have an attitude of ‘let him be, we understand, come here, we’ll get along*”.

The process of connectedness is intensified around a joint music activity which results in a collective musical product, perceived as positive and attractive by the children/adolescents. A teacher said: “*The fact that children can make themselves heard in a way without the need for language. With an instrument. I think that is very strong and that children also get control. They play something on an instrument and the other children clap it. Then you see children really grow, hey, children really like that too. I think that’s a really successful experience.*”. Another teacher mentioned: “So, each child has his own input in the song and together that is a group whole. So, we are responsible for that song together as a group. Mmh, and everyone has their own voice in that. Yes, I always like that very much.” Additionally: “*To make that song together at the end, yes, I find that a magical moment, a very, yes, a very beautiful part. That connects the group, and it results in an end product in which everybody can be his own ‘self’ in that, and show that, let themselves be heard.*” (teacher).

#### 3.2.2. Process Result: Sense of Belonging

The results of the “process of strengthening social connectedness” can be summarized as a “*sense of belonging*”, which refers to the psychological feeling of belonging or connectedness to a social group or a community [30,31]. Teachers and music therapists describe that children/adolescents show feeling connected to the group, feeling seen and heard, feeling accepted and valued, and display emotions of pride and self-esteem. A participating teacher reported, ”*I noticed that when they listen to the song they have recorded, they recognize each other and give each other more compliments than they do regularly. They are conscious that they are a group and the song is from them as a group.*”. A participating music therapist reported, “*I did not expect that the song did so much, I noticed that it created a sense of belonging. The class started spontaneously with a small hand game together, I‘ve never seen that before*”.

A teacher said the following: “I thought that it already was a fairly close group, but yes a few children who then fall out a bit, but yes I do think it has strengthened them somewhat in the group feeling and I certainly felt that when they were composing that song. Then, you noticed that, yes that there was a good flow and cooperation between the students, yes.”. One of the music therapists mentioned signals of self-esteem in one girl: “There were also students who discovered their talents, for example a girl who didn’t really know what her talent was, but writing was a talent, she said as she went along in the process. And in the end, she even rapped and took the lead and suggested making a music video for it. Yes, it was as if she, how do you say, blossomed.”

A music therapist reported results on solidarity within the group: “But maybe it is for them that sense of solidarity and be a part of something for the first time in a new land, so they see it from a different perspective”. A teacher mentioned: “So, that it seems as if they look out for each other a little bit that they are a bit like well leave him but yes, we understand that come on here, we’ll take you under our arms. In the class of (says teacher’s name) of this group yes, yes and I think that’s nice behavior, almost protective uhm, what parents also do with their children.”. 

In addition, teachers and music therapists reported that the intervention was successful regardless of language, musicality, age, or cultural background. In that regard, the intervention stimulates inclusiveness in the classroom. One teacher mentioned: “*Also the most arrhythmic one, can still join*”. Another teacher said “*Children who are actually very timid, do get a chance to express themselves and to show something. And that, I think is the beauty of it. That is*
*more than in a lesson situation where they are constantly judged on whether you say that sentence correctly in Dutch. Nothing is judged there at all, so you give them room to come out of that shell and I think that’s very beautiful”.*

Second, the intervention provides room for cultural differences and teachers reported the stimulation of a culturally sensitive approach. One teacher said: “that moment I started looking for the music of her country I saw a revival”, and a music therapist mentioned, “*Uhm, so they’re really looking for that connection. Of course I also searched with them for what is a theme that connects you? And then we colored that with things like cultures, respect for each other, what do you think is important about friendship, what is valuable? What do you find difficult*?”.

Teachers mentioned that the interventions support supervision in order to create a “sense of belonging” and a sound learning climate in the classroom (see also Figure 1). One quote of a teacher illustrates this: “*Now that I have seen how he (music therapist) has tackled that and how the children react to that, I think yes, that is a completely different approach than from a school perspective*”. Teachers also reported that the intervention contributes to the adaptation to specific needs of refugee children/adolescents: “*I also like to include the findings of the music therapist if you want to map the behavior of a child. To take him there as an outsider with a different view on the starting point and to ask for advice*”.

Finally, music therapists stated that the clear structure and progressive building of the intervention can contribute to building a safe environment. 

#### 3.2.3. Influencing Factors

The process of social connectedness and its results can be strengthened or hampered by the following factors: the initial situation regarding the level of social connectedness or learning climate in the group (at the start of the intervention); the possibility of meeting the prerequisites of the intervention; the professional competence and experience of the music therapist; and the amount of collaboration between all professionals involved.

Initial situation

The initial situation describes the initial level of social connectedness or climate in the group during the start of the intervention. The initial situation is related to the extent and the type of psychosocial problems of the individual children/adolescents; the diversity of cultures and personalities; the extent to which children/adolescents feel safe to communicate and express themselves; the age category of the group; the competence to understand the instructions given; the group size; and the stability of the group. Major differences in age, as well as a diversity in cultures and/or intellectual level within the group might be a barrier for adequate adaptation of the intervention in the group. In addition, high levels of social connectedness in the group at the start of the intervention were related to a smaller range to improve connectedness or climate. A participating teacher remarked: “*I think it was already a relative close group with a few people who were socially excluded to some extent. I think it has strengthened them in their sense of belonging to a group*”. Another teacher mentioned: “*In this group I think it plays a part that the mutual relationships are very disturbed anyway and it is a younger group so you notice that as well in the music session.*”.

Possibility to meet prerequisites of the intervention

The prerequisites for the intervention were the number of group sessions, the duration of the sessions, the embedding of the music therapist in the group, and whether the instructions and indications for additional individual sessions were clear. 

Teachers reported that the eligibility criteria for children/adolescents to participate in the individual sessions were not clear. The ambiguity and lack of transparency in the communication regarding the access to the individual sessions, to children/adolescents, parents, and other professionals were reported as a point for improvement. A participating teacher remarked, “*Sometimes it is difficult to explain why one child can get individual sessions and the other one doesn’t. They all could use it.*”.

Furthermore, the music therapists reported that the number and duration of the group sessions during this study were insufficient. Several music therapists experienced that, due to the limited time spent in the group, they were not able to support the process sufficiently. They expected that a more structural and prolonged presence in the group would contribute to building relationships within the group. A participating music therapist responded: “*I find it difficult what to expect in a few sessions, so you cannot expect a lot. That is a difficult one, what can you do in such a short period?*”.

Professional competence and experience of musical therapist

Conducting the intervention requires certain competences of the music therapists, such as the ability to adapt the themes and activities to the age of the children/adolescents, the climate in the group, and current events that have taken place in the group. Two music therapists reported that gaining experience in conducting the intervention is important in order to be able to tailor the intervention to the group. A participating music therapist reported, “*At the youngest group I brought up the themes talents, emotions and difficult events. But they just didn’t understand me. Difficult events did not concern them. So I had to go back to very basis programs.*”: This is also a topic which was discussed during the intervision between the music therapists. Safe & Sound consists of three phases to build up safety. If the therapist is working too fast, children have problems in understanding the therapy and the music therapist has to take one step back.

The collaboration between professionals involved

The collaboration between the professionals involved concerns the communication between the music therapists, parents, and teachers. It is essential that involved teachers and music therapists meet on a regular basis in order to monitor changes in the learning climate and in (the behavior of) individual children/adolescents. In daily practice, the opportunity for mutual consultation is limited due to restrictions in time. The communication between these professionals is related to the initial situation, to choices regarding appropriate and appealing themes, to the division of tasks and roles, and to the communication with parents and stimulating them to participate. A participating teacher remarked, “*One moment I felt I had to intervene in the group but then I thought: the music therapist is working and I don’t want to be in the wheels of the therapist. Yes, we didn’t make clear agreements about this before we started.*”. A participating music therapist remarked, “*When I look back, I see I didn’t make clear agreements beforehand with the teachers. I think I needed more communication with the teachers, but I couldn’t have done it differently because I know more now. So if I had to do it again, I would have done it differently, but some things I didn’t know when I started. I didn’t know it was my task in the beginning. Now I would have communicated with the teachers every two to three weeks*”.

### 3.3. Quantative Results

Prior to analysis, scores for all variables were examined for accuracy of data entry, distributions, univariate, and multivariate outliers. Two cases with extremely high z-scores (*z* > 3.29) on negative affect (T_2_) and one case with extremely low z-scores (*z* < −3.29) on atmosphere (T_0_ and T_1_) were found to be univariate outliers. The case with two univariate outliers on atmosphere T_0_ and T_1_ was also identified as a multivariate outlier based on the Mahalanobis distance (*p* < 0.001). All three cases containing the univariate outliers were removed from the analysis. Missing values were found in all items and were imputed with predictive mean matching based on all research variables. The completed data from the research variables are summarized in Table 2. Shapiro-Wilk’s tests for normality of distributions indicated that all variables other than negative affect at T0 deviated from the normal distribution and were positively skewed (*p* < 0.001).

#### 3.3.1. Negative Affect

A hierarchical multilevel regression analysis was performed on negative affect. The addition of time (model 2) as a predictor significantly improved upon the null-model (model 1), (*Δdeviance*(1) = 4.977, *p* = 0.026). The random slope for time was not a significant improvement of model 3 over model 2 (*Δdeviance*(1) = 1.516, *p* = 0.218). The addition of school-type did not significantly improve the model (*Δdeviance*(1) = 0.272, *p* = 0.602), nor was it a significant predictor of negative affect (Model 4) (*B* = 0.200, *t*(106) = −0.735, *p* = 0.464). Both model 3 and model 4 had higher AIC than model 2. For these reasons, model 2, a growth model with only a random intercept, was chosen as the “final” model. In model 2, negative affect decreased significantly over time (*B* − 0.31, *t*(215) = −3.18, *p* = 0.002, see Table 3). 

#### 3.3.2. Classroom Climate

A hierarchical multilevel regression analysis was performed on six different aspects of classroom climate: support, growth, repression, atmosphere, environment, and safety. Only support was found to significantly change over time; all the other classroom climate variables were not significantly different over time (see Table 4). Except for support, the addition of time as a predictor, either with random-intercept only, or random intercept and slope did not significantly improve model fit (Δdeviance(1) < 3.841, *p* > 0.05) or result in higher AIC-scores, and in none of these models was time found to be a significant predictor of classroom climate. 

For support, the addition of time (model 2) as a predictor with only a random intercept did not significantly improve upon the null-model (model 1), (Δdeviance(1) = 2.329, *p* = 0.127). However, allowing for random slopes for time significantly improved model 3 over model 2 (Δdeviance(1) = 4.680, *p* = 0.031), and was the model with the lowest AIC of all four models. The addition of school-type did not significantly improve the model (Δdeviance(1) = 0.607, *p* = 0.436), nor was it a significant predictor of support (model 4) (*B* = −0.239, *t*(106) = −1.137, *p* = 0.258). Because of the lowest AIC found in model 3 and its significant improvement over model 2, model 3, a growth model with random intercept and slope for time was chosen as the “final” model. In model 3, support decreased significantly over time (*B* = −0.17, *t*(215) = −2.099, *p* = 0.037, see Table 4).

### 3.4. Treatment Integrity 

The music therapists followed the instruction manual most of the time. For the group sessions, the mean score was 92% (range over the session 50–100%) and for the individual sessions this was even higher, i.e., 99% (range over the session 80–100%). No therapeutic techniques were added in the individual sessions. In the group sessions, techniques were added in nearly 80% of the sessions. The music therapist felt competent to apply the intervention (95% for the group sessions and 97% for the individual sessions). In nearly all sessions, music therapists applied reflective questions on emotions, i.e., in 85% of the group sessions and in 88% of the individual sessions. Questions on talents were asked in about half of the sessions, 51% and 57%, respectively.

## 4. Discussion 

The present mixed-method study aimed to set the first steps in the implementation of the music therapy intervention “Safe & Sound” and to get insight in the process of implementation, its process results, influencing factors, and the perceived effects of the intervention in refugee children and adolescents. The results of the process evaluation indicated that the intervention Safe & Sound strengthens the process of social connectedness, resulting in a “sense of belonging”, which refers to the psychological feeling of belonging or being connected to a social group or a community [41]. This can be interpreted as an early stage of resilience [42]. This was reported during the interviews by both teachers and music therapists. Teachers reported that the intervention can stimulate inclusiveness in schools and culturally sensitive attitudes. The quantitative results indicated that refugee children and adolescents showed a decrease of negative affect during the time of the intervention and is in line with the qualitative results of teachers and music therapists in this respect. It is plausible to suggest that the Safe & Sound intervention stimulated a number of protective factors for the children in the classroom, such as maintenance of cultural identity, social support, belonging, and safety. These factors are protective for mental health problems and psychological functioning in refugee children [42]. This was also seen in our study since children and adolescents showed a decrease over time in negative affect. Peer support, the education system, and “acculturation” have been shown to be protective factors in especially positive adjustment [42,43]. 

The results of the interviews further indicate that teachers mentioned that Safe & Sound helped them in creating a sound learning climate and to better adapt to the specific needs of refugee children. Music therapists stressed the contribution of the intervention to a safe environment because of the clear structure and progressive phasing of the intervention. These beneficial results on the classroom climate were not seen in the experience of the refugee children and adolescents. They did not experience a better classroom climate, such as a better atmosphere or more safety in the classroom. However, ceiling effects were seen on all variables regarding classroom climate and on all measurement time points. Even for one element of the classroom climate, i.e., support of the teacher, a decrease was seen over time. This could possibly be explained by contextual factors (Christmas holidays, changes in constitution of the children in the class), but also by the fact that results proceed a regression to the mean as high scores at the first measurement will likely decrease and be closer to its mean. It should also be taken into account that the description of this item, “support”, refers to teachers and not directly to the intervention itself. 

A second aim was to identify influencing factors for the implementation of the intervention. Results have shown that the initial situation in the classroom, the possibility of meeting the prerequisites of the intervention, the professional competence and experience of music therapists, and the collaboration and communication between the involved professionals can be seen as influencing factors. First, the level of social connectedness or learning climate in the group at the start of the intervention was highly influenced by the diversity of cultures, age, language and personalities, feelings of safety, the size, and the stability of the group. Major differences in age, as well as a diversity in cultures and/or intellectual level within the group might be a barrier for adequate adaptation of the intervention in the group. Second, the possibility of meeting the prerequisites of the intervention was mentioned as an important factor in the process. In this regard, the importance of equivocal and transparent communication by and between teachers and music therapists were mentioned, in particular, regarding the access to the individual sessions. Additionally, the number and duration of group sessions were seen as insufficient. Results showed the importance of a structural and prolonged presence of the therapists in the group. This would contribute to building stronger relationships within the group. This is in accordance with a dose–response relationship that was previously found in music therapy [32]. Third, conducting the intervention requires certain competences of the music therapists, such as the ability to tailor it to the specific needs of the group and its heterogeneity. The music therapists that conducted the intervention for the first time reported sufficient treatment integrity and felt competent enough. Nevertheless, during some sessions the scores were lower (<60%), which was explained by music therapists by difficulties in proper tailoring to the group, events, or individuals in the group. Training and discussing these themes during intervision were seen as very helpful. Last, it is seen as essential that teachers, music therapists, and parents are adequately aligned on the psychological functioning and mental health of the child or adolescent. In daily practice, the opportunity for mutual communication between teachers and music therapists is limited due to restrictions in time.

These influencing factors lead to several recommendations for implementation of this (or a comparable) intervention in the future. First, it is important to make room and time available for communication between teachers and music therapists. Music therapists could join the weekly meetings of the teachers or a specific meeting on mental health could be introduced. Second, before starting the intervention, it is desirable that the music therapist is introduced to the functioning of the class in order to get insight in the initial situation. Third, it is recommended to elaborate on the specific eligibility criteria for the individual session. Fourth, the training and intervision process could be strengthened, so music therapists are better able to tailor the intervention. Lastly, the number of sessions implemented could be more flexible and, if needed, more sessions could be offered.

### Strength and Limitations

There are some methodological considerations. A strength is this mixed-method design, which made it possible to get insights from different perspectives, i.e., teacher, music therapist, children, and adolescents. The qualitative research appeared to be a successful method to investigate an area where research is still sparse [44]. The strength of the qualitative results lies in the richness of experiences as expressed by the participants. 

The sampling strategy can be characterized as convenience sampling [44]. All participating music therapists and teachers were involved in the data collection. Working with a small number of participants in the interviews enabled us to examine their responses in depth. The actual research started the analysis with the interviews with the teachers. Data saturation occurred after about half of the interviews with the music therapists, which means that no new categories emerged. The qualitative analysis of the last interviews served to confirm the results. This process that was seen in this study was context-dependent, so it is important to replicate this study in other schools. The present study provides initial insights into the implementation process. 

Further limitations of this study were the sample and the lack of a control group, which makes it impossible to exclude other factors than the intervention when interpreting the quantitative results. Therefore, this study needs to be considered as a first step in studying Safe & Sound among refugee children. Future studies should focus on further implementation of Safe & Sound in order to increase the generalizability, as well as investigating the feasibility of the study with a proper control group. Additionally, multiple single-case experimental designs on the individual sessions should be planned as a next step in order to get more insights into the effects of the individual part of the intervention. 

## 5. Conclusions

This study presents the first results of the implementation of the intervention Safe & Sound. Safe & sound has a resilience-focused approach and includes elements that have been shown to be protective for mental health in refugees. When implementing the intervention in schools, it is important to take into account the initial situation in the classroom, the prerequisites for the intervention, the professional competence, the experience of music therapists, and the collaboration and communication between the professionals involved. Future research should shed more light on the effects of further implementation of music therapy, especially on the longitudinal protection of trauma-based problems in refugee children. 

## Figures and Tables

**Figure 1 children-09-01434-f001:**
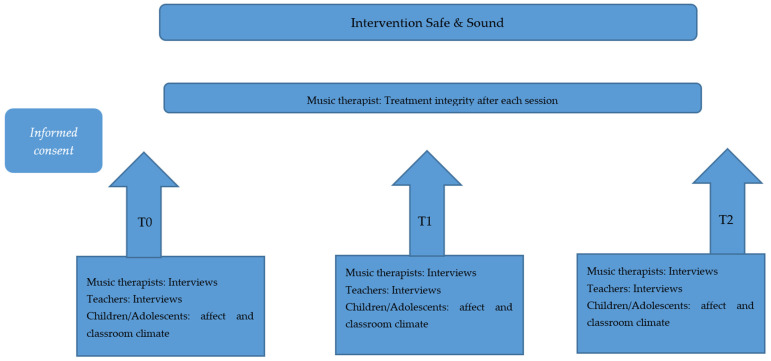
Procedure.

**Figure 2 children-09-01434-f002:**
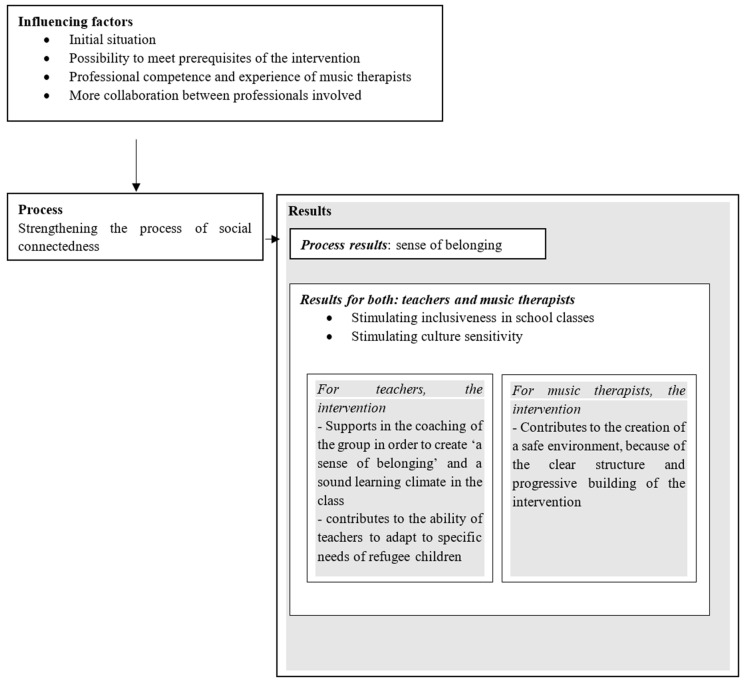
Process and process results of the interventions Safe & Sound.

**Table 1 children-09-01434-t001:** Fit measures and McDonald’s omega for the VAS variables positive and negative affect.

		χ^2^(df)	CFI	RMSEA	SRMR	ω Positive Affect	ω Negative Affect
Model 1 (all items)	T0	54.39(19) ***	0.737	0.131	0.089	0.442	0.592
	T1	49.10(19) ***	0.816	0.121	0.085	0.453	0.599
	T2	68.64(19) ***	0.715	0.156	0.100	0.438	0.605
Model 2 (without “easily angered”)	T0	45.76(13) ***	0.754	0.153	0.088	0.450	0.663
	T1	39.84(13) ***	0.832	0.138	0.083	0.444	0.683
	T2	60.09(13) ***	0.713	0.183	0.105	0.450	0.629

*** = *p* < 0.001.

**Table 2 children-09-01434-t002:** Means and standard deviations of VAS-negative affect and classroom climate.

	T_0_		T_1_		T_2_	
	*Mean*	*SD*	*Mean*	*SD*	*Mean*	*SD*
Negative Affect	2.08	1.98	2.18	1.86	1.47	1.50
Support	9.32	1.09	8.82	1.71	8.99	1.46
Growth	9.15	1.25	9.13	1.15	9.12	1.20
Atmosphere	8.32	2.13	8.24	2.21	8.29	2.05
Environment	8.44	2.02	8.51	1.56	8.36	2.12
Repression	8.83	2.05	8.12	2.38	8.37	2.13
Safety	8.76	1.95	8.81	1.70	9.06	1.63

**Table 3 children-09-01434-t003:** Unstandardized coefficients and model fit indices for the four multilevel models predicting negative affect. Model 1 = intercept only, Model 2 = random intercept for time, Model 3 = random intercept and slope for time, Model 4 = random intercept for time. Level 1 = time, level 2 is student.

		Model 1			Model 2			Model 3			Model 4		
		B	SE	*p*	B	SE	*p*	B	SE	*p*	B	SE	*p*
Negative Affect	Intercept	1.911	0.133	<0.001	2.522	0.233	<0.001	2.522	0.253	<0.001	2.599	0.256	<0.001
	Time				−0.305	0.096	0.002	−0.305	0.097	0.002	−0.305	0.096	0.002
	School-type										−0.2	0.272	0.464
	Deviance	−632.144			−627.167			−625.65			−626.895		
	AIC	1270.288			1262.334			1263.301			1263.789		
	*p* Δdeviance				0.026 ^1^			0.218 ^2^			0.602 ^3^		

NOTE: ^1^ Model 2—model 1, ^2^ model 3—model 2, ^3^ model 4—model 2.

**Table 4 children-09-01434-t004:** Unstandardized coefficients and model fit indices for the four multilevel models for each classroom climate construct. Model 1 = intercept only, Model 2 = random intercept for time, Model 3 = random intercept and slope for time, Model 4 = random intercept for time. Level 1 = time, level 2 is student.

		**Model 1**			**Model 2**			**Model 3**			**Model 4**		
		**B**	**SE**	* **p** *	**B**	**SE**	* **p** *	**B**	**SE**	* **p** *	**B**	**SE**	* **p** *
Support	Intercept	9.045	0.106	<0.001	9.383	0.189	<0.001	9.383	0.167	<0.001	9.475	0.187	<0.001
	Time				−0.169	0.078	0.032	−0.169	0.08	0.037	−0.169	0.08	0.037
	School-type										−0.238	0.209	0.258
	Deviance	−561.326			−558.998			−554.318			−553.711		
	AIC	1128.653			1125.996			1120.635			1121.422		
	*p* Δdeviance				0.127 ^1^			0.031 ^2^			0.436 ^4^		
		**Model 1**			**Model 2**			**Model 3**			**Model 4**		
		B	SE	*p*	B	SE	*p*	B	SE	*p*	B	SE	*p*
Growth	Intercept	9.133	0.085	<0.001	9.16	0.159	<0.001	9.16	0.163	<0.001	9.263	0.177	<0.001
	Time				−0.014	0.067	0.837	−0.014	0.07	0.842	−0.014	0.07	0.842
	School-type										−0.263	0.173	0.131
	Deviance	−502.511			−502.489			−502.232			−501.081		
	AIC	1011.021			1012.978			1016.464			1016.162		
	*p* Δdeviance				0.884 ^1^			0.612 ^2^			0.235 ^3^		
		**Model 1**			**Model 2**			**Model 3**			**Model 4**		
		B	SE	*p*	B	SE	*p*	B	SE	*p*	B	SE	*p*
Atmosphere	Intercept	8.284	0.155	<0.001	8.312	0.279	<0.001	8.312	0.279	<0.001	8.705	0.299	<0.001
	Time				−0.014	0.116	0.905	−0.014	0.116	0.905	−0.014	0.116	0.905
	School-type										−1.01	0.303	0.001
	Deviance	−684.236			−684.229			−684.228			−678.881		
	AIC	1374.472			1376.457			1380.455			1367.762		
	*p* Δdeviance				0.932 ^1^			0.973 ^2^			0.021 ^3^		
		**Model 1**			**Model 2**			**Model 3**			**Model 4**		
		B	SE	*p*	B	SE	*p*	B	SE	*p*	B	SE	*p*
Environment	Intercept	8.438	0.135	<0.001	8.522	0.254	<0.001	8.522	0.25	<0.001	8.693	0.273	<0.001
	Time				−0.042	0.108	0.699	−0.042	0.114	0.715	−0.042	0.114	0.716
	School-type										−0.44	0.273	0.111
	Deviance	−654.281			−654.206			−652.434			−651.154		
	AIC	1314.563			1316.412			1316.868			1316.307		
	*p* Δdeviance				0.784 ^1^			0.183 ^2^			0.081 ^3^		
		**Model 1**			**Model 2**			**Model 3**			**Model 4**		
		B	SE	*p*	B	SE	*p*	B	SE	*p*	B	SE	*p*
Repression	Intercept	8.441	0.146	<0.001	8.904	0.303	<0.001	8.904	0.289	<0.001	8.897	0.325	<0.001
	Time				−0.231	0.133	0.082	−0.231	0.134	0.085	−0.231	0.133	0.083
	School-type										0.018	0.3	0.952
	Deviance	−708.66			−707.136			−706.333			−707.134		
	AIC	1423.319			1422.271			1424.666			1424.268		
	*p* Δdeviance				0.217 ^1^			0.370 ^2^			0.966 ^3^		
		**Model 1**			**Model 2**			**Model 3**			**Model 4**		
		B	SE	*p*	B	SE	*p*	B	SE	*p*	B	SE	*p*
Safety	Intercept	8.877	0.118	<0.001	8.58	0.243	<0.001	8.58	0.255	<0.001	8.588	0.273	<0.001
	Time				0.148	0.106	0.164	0.148	0.110	0.181	0.148	0.11	0.181
	School-type										−0.019	0.242	0.937
	Deviance	−636.887			−635.909			−635.407			−635.404		
	AIC	1279.775			1279.818			1282.814			1284.807		
	*p* Δdeviance				0.323 ^1^			0.479 ^2^			0.477 ^3^		

^1^ Model 2—model 1, ^2^ model 3—model 2, ^3^ model 4—model 2, ^4^ model 4—model 3.

## Data Availability

The data presented in this study are available on request from the corresponding author.

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
