# Peer review of "A Music Therapy Intervention for Refugee Children and Adolescents in Schools: A Process Evaluation Using a Mixed Method Design"

_children, 2022, doi:10.3390/children9101434_

Round 1
Reviewer 1 Report
Review
First of all, I would like to report that the article seems to me to be of great interest to the scientific community. Next, I will describe certain issues that will allow the authors to improve the study.
With respect to the theoretical framework, it is necessary to refer to music therapy interventions and the results of different studies where this therapy is used in children and adolescents at risk of social exclusion in case we do not find material referring specifically to refugees. I reiterate that it would also be necessary to approach the subject more deeply at a theoretical level in order to understand the possibilities of music as a therapeutic tool.
At the level of methodological design it is necessary to explain the instruments precisely, both what the questionnaires consisted of with their reliability index and the thematic blocks of the interviews and other issues used, as presented is scarce. It is also necessary to clarify what the music therapy interventions consisted of, not only whether they were joint or individual, but also the structure of the sessions.
With respect to the results, the classification by qualitative or quantitative instruments is confusing; it would be necessary to present the data by study variables in such a way as to allow triangulation of the data and to present them in a clearer way. This section should be redone, as the presentation of the data and results is confusing.
In the conclusions section, it would also be interesting to include some future lines of action to open up new research in the short term.

Author Response
see attachement

Reviewer 2 Report
The study is very interesting and deals with a current and very important topic. Particularly positive is the mixed-method approach, which clarifies the inclusion of different perspectives.
If a few more details were added to the text, I think it would gain in quality.
"The current research was conducted as a mixed-methods longitudinal study, in which both quantitative as well as qualitative methods were applied" (p.76/77)
--> What kind of mixed-method design has been applied? Parallel Design, Embedded Design? Please add reference e.g. Creswell & Plano-Clark
"If children/adolescents were considered for the individual sessions, parents were informed and asked for their approval" (p.146/147)
--> What are the selection criteria?
Some open questions:
Was there a training for the Safe and Sound intervention?
Was accompanying intervision or supervision obligatory?
Could you provide more information about the instruction manual?
The analysing methods are state of the art and very well described (especially the different R packages).
Results
"50 boys and 50 girls were included, for nine participants gender 294 information was missing." (p.294) --> 108 in total, so is it for eight participants?
p.346 ff
It is great to read the quotes of the interviews. For comprehensibility, it would be helpful to add line details.
In the discussion it would be very exciting to get a deeper interpretation of the models presented in the results. How are negative B-values in the models to be interpreted?
Author Response
see attachement
